# Endoscopic Ultrasonography Diagnosis of Early Pancreatic Cancer [note 1]

**DOI:** 10.3390/diagnostics10121086

**Published:** 2020-12-14

**Authors:** Keisuke Kurihara, Keiji Hanada, Akinori Shimizu

**Affiliations:** Department of Gastroenterology, Onomichi General Hospital, 1-10-23, Hirahara, Onomichi 722-8508, Japan; kh-ajpbd@nifty.com (K.H.); a.shimizu313@gmail.com (A.S.)

**Keywords:** endoscopic ultrasonography, pancreatic cancer, early diagnosis

## Abstract

Early diagnosis of pancreatic cancer (PC) can improve patients’ prognosis. We aimed to investigate the utility of endoscopic ultrasonography (EUS) for the early diagnosis of PC. This study included 64 patients with PC at an early stage treated at Onomichi General Hospital between January 2007 and January 2020. Diagnostic procedures included contrast computed tomography (CT), magnetic resonance cholangiopancreatography, EUS fine-needle aspiration, and endoscopic retrograde cholangiopancreatography (ERCP) for pancreatic juice cytology. The mean age was 71.3 years. In all, 32 patients were stage 0, and 32 were stage I. As for image findings, the main pancreatic duct (MPD) stenosis was detected in several cases, although CT and MRCP seldom detected tumors. EUS had a high detection rate for stage 0 tumor lesions. The median observation period was 3.9 years. In cases with stage 0, the 1 year and 5 year survival rates were 100% and 78.9%, respectively. In cases with stage I, the 1 year and 5 year survival rates were 96.4% and 66.7%, respectively. EUS has the highest sensitivity among all imaging modalities for detecting small pancreatic tumors. Cases with MPD dilation or stenosis, especially with tumors that cannot be identified on CT and MRI, should have EUS performed. In some cases, EUS was not able to detect any tumor lesions, and ERCP-based pancreatic juice cytology should be useful for pathological diagnosis.

## 1. Introduction

According to the Vital Statistics of Japan reported by the Ministry of Health, Labor, and Welfare [1]), 40,981 patients were diagnosed with pancreatic cancer (PC) in 2017, and the number is increasing yearly. In 2018, 35,390 patients died of PC, and the mortality rate was 29.7 per 100,000 men and 27.4 per 100,000 women. PC has the fourth highest mortality rate of all cancers in Japan, and the 5 year survival rate is 8.7%. The poor prognosis is attributed to the difficulty in diagnosing PC at an early stage; it is usually diagnosed at an advanced stage [2,3]. However, when PC is diagnosed at an early stage, it has a good prognosis. According to an analysis from the Japan Pancreatic Cancer Registry, the 5 year survival rates of patients with Union for International Cancer Control (UICC) stage IA and stage 0 were 68.4% and 85.8%, respectively [4].

We developed an initiative for the early detection of PC, which involved collaboration between PC specialists from medical centers and general practitioners [5]. The specialists used endoscopic ultrasonography (EUS) in addition to contrast computed tomography (CT) and magnetic resonance imaging (MRI). EUS is an ultrasound technique in which the tip of the endoscope is equipped with a high-frequency transducer. EUS has a high resolution, and there are many reports of its high sensitivity for detecting PC. In this study, we aimed to investigate the clinical features of early-stage PC and the utility of EUS for diagnosing early-stage PC. 

## 2. Material and Methods

### 2.1. Early Diagnosis Initiative for PC (the Onomichi Project)

In 2007, the Onomichi Medical Association created a social program in which PC specialists from medical centers collaborated with general practitioners to improve the early diagnosis of PC [5]. First, the specialists educated general practitioners about the risk factors of PC (Table 1). The importance of elevated serum levels of pancreatic enzymes, ultrasound (US) findings such as pancreatic duct dilation, and the importance of magnetic resonance cholangiopancreatography (MRCP) and EUS in the detection of early-stage PC were discussed. If general practitioners encountered such cases, they referred patients to the specialists to check for possible PC (Figure 1).

### 2.2. Patient Investigations

We conducted a retrospective study, including patients who had undergone surgery and were diagnosed with PC from January 2007 to January 2020 at Onomichi General Hospital. The number of patients who consulted for a suspected PC was 18507 cases, while that of patients who underwent CT was 8576. Additionally, 7492 patients who had main pancreatic duct (MPD) dilation underwent MRCP. Based on the CT and MRCP results, tumor lesions, irregular MPD stenosis, pancreatic duct branch dilation, and a pancreatic cyst were detected; thus, we performed EUS on 4037 patients. If a tumor lesion was detected and had a considerable size for fine-needle-aspiration (FNA), we performed EUS–FNA to diagnose 633 patients. In cases of negative FNA, tumor lesion not detected with EUS, or lesions too small for FNA, we performed endoscopic retrograde cholangiopancreatography (ERCP) for pancreatic juice cytology on 780 cases. A total of 610 cases were histologically diagnosed with PC, while 290 underwent surgery. Sixty-four cases of early-stage PC were investigated postoperatively, consisting of 32 cases with stage 0 and 32 cases with stage I (Figure 2). We performed EUS for all suspicious PC cases. However, EUS was not performed in patients who refused or cases before 2010 when EUS was not commonly performed in hospitals. We investigated the clinical features of the cases and the utility of EUS for diagnosis. EUS was performed using a radial echoendoscope (GF-UE260 and GF-UE290, Olympus Medical Systems, Tokyo, Japan) equipped with a processor (UE ME-1 and UE ME-2, Olympus Medical Systems). EUS was performed by experienced specialists with more than 5 years of working experience and performed an average of more than 200 procedures every year.

### 2.3. Risk Factors for PC

In Japan, the clinical guidelines for PC based on evidence-based medicine were published by the Japan Pancreas Society. In the clinical question session, risk factors for PC were suggested (Table 1) [6]. According to the fifth version in 2019, and the diagnostic algorithm recommends that EUS should be performed by an experienced endoscopist who is proficient in the EUS technique. They also recommend EUS as a diagnostic tool in subjects with suspected PC because it is more sensitive than other imaging modalities [7]. However, the indication for EUS should be carefully determined because it may be a relatively invasive procedure [8]. 

## 3. Results

### 3.1. The Clinical Characteristics and Imaging Findings of Stage 0 and Stage I PC 

This study included 64 patients who underwent surgery at Onomichi General Hospital between January 2007 and January 2020. Of these, 32 patients were classified as UICC stage 0, including 23 pancreatic cancer in situ (PCIS), and nine high-grade dysplasia from intraductal papillary mucinous neoplasm, and 32 patients as stage I. The mean ages of patients with stage 0 and stage I were 71.3 (range, 52–87) years and 71.8 (range, 39–84) years. There were 19 men and 13 women with stage 0 and 14 men and 18 women with stage I. 

Stage 0 PCs were located in the pancreatic head in 11 patients, the body in 16, and tail in five. For stage I PC, it was 14, 16, and two, respectively. The median observation periods were 4.2 years and 3.4 years in stage 0 and stage I, respectively. For patients with stage 0, the 1 year and the 5 year survival rates were 100% and 78.9%, respectively. For stage I, the 1 year and 5 year survival rates were 96.4% and 66.7%, respectively (Table 2).

We investigated EUS, CT, and MRI detection rates for MPD stenosis, MPD dilation, and tumor lesions. EUS was performed on 22 and 22 patients with stage 0 and stage I, respectively. CT was performed on 31 and 27 patients with stage 0 and stage I, respectively. MRI was performed on 31 and 23 patients with stage 0 and stage I, respectively. 

For stage 0, MPD stenosis was detected in 16 (72.7%), 14 (45.2%), and 18 (58.1%) patients by EUS, CT, and MRI, respectively. For stage I, this was detected in 17 (77.3%), 15 (55.6%), and 15 (68.2%) patients by EUS, CT, and MRI, respectively. 

For stage 0, MPD dilation was detected in 17 (81%), 19 (61.3%), and 20 (64.5%) patients by EUS CT, and MRI, respectively. For stage I, this was detected in 16 (72.7%), 18 (66.7%), and 18 (78.3%) patients by EUS, CT, and MRI, respectively. 

For stage 0, tumor lesions were detected in 10 (45.5%), 3 (9.7%), and 3 (9.7%) patients by EUS, CT, and MRI, respectively. For stage I, these were detected in 18 (81.8%), 15 (63%), and 9 (39.1%) by EUS, CT, and MRI, respectively (Table 3). 

### 3.2. EUS Imaging Findings of Stage 0 PC 

We detected tumor lesions in 10 cases of stage 0 PC. All cases had MPD stenosis. Low echoic round lesions ranging from 10 to 20 mm in diameter were detected around the stenosed pancreatic ducts. 

Six cases showed well circumscribed low echoic lesions as a tumor mass (Figure 3a). The other four cases showed a pale, low-echoic lesion with a relatively circumscribed lesion and could be overlooked if not careful (Figure 3b). It is not certain what factors cause the differences in these features. Some cases of stage 0 had a low echoic lobular background, the result of chronic pancreatitis, and in such cases, it could be difficult to detect the tumor. 

### 3.3. Diagnosis of Early-Stage PC 

For stage 0, although EUS detected tumor-like lesions in some cases, they were too small for EUS–FNA, and all patients underwent pancreatic juice cytology after ERCP. We inserted a 0.025 inch guidewire into the pancreatic duct and introduced a 4 or 5 Fr endoscopic nasopancreatic drainage (ENPD) catheter (Gadelius Medical, Tokyo, Japan) into the MPD. Pancreatic juice can be collected at five or six times from an ENPD tube (Figure 4). The sensitivity of pancreatic juice cytology was 87.5% (28/32). Five patients with cytology-negative findings were suspected to have PC based on the imaging results and underwent surgery after providing consent; all cases were diagnosed as stage 0 PC. For stage I, 18 cases (81.8%) had tumor lesions detected with EUS, and EUS–FNA was performed in 12 cases. The sensitivity of EUS–FNA was 83.3% (10/12). Two cytology-negative cases diagnosed by EUS–FNA underwent pancreatic juice cytology after ERCP, and cytology showed positive as adenocarcinoma. A total of 22 cases underwent pancreatic juice cytology in stage I, and the sensitivity was 81.8% (18/22). 

## 4. Discussion

### 4.1. The Utility of EUS in the Detection of Early-Stage PC

EUS is now the major modality for the detection of pancreatic lesions. From a large number of studies (*n* = 1170), the median sensitivity of EUS for detecting pancreatic tumors was 94%. It had a higher sensitivity than CT and MRI [9]. EUS has a high resolution and can detect small lesions [10]. Muller et al. examined 49 pancreatic tumors smaller than 30 mm and reported EUS, CT, and MRI sensitivities as 93%, 53%, and 67%, respectively [11]. Maguchi et al. reported that for tumors smaller than 2 cm in diameter, except for carcinoma in situ, the detection rate of EUS was as high as 100% (9/9), in contrast to 60% (6/10) and 50% (5/10) for US and CT, respectively [12]. Sakamoto et al. also reported that for tumors smaller than 2 cm, the sensitivity of EUS was higher than that of contrast-enhanced CT (94.4% vs. 50%, respectively; *n* = 36) [13]. Several reports have shown that EUS could detect pancreatic tumors that were not identified with other modalities [7,14,15]. According to an analysis by the Japan Study Group on the Early Detection of Pancreatic Cancer (JEDPAC), of the imaging findings of 51 cases of stage 0 and 149 cases of stage I PC, the most common finding was MPD dilatation. The detection rates of CT, MRI, and EUS were 79.6%, 82.7%, and 88.4%, respectively. The sensitivities for detecting tumor lesions on CT scan, MRI, and EUS imaging for stage 0 were 10%, 10.9%, and 24.4%, respectively, and for stage I were 65.8%, 57.5%, and 92.4%, respectively [16]. 

Among these results, EUS has the highest sensitivity of among the imaging modalities for detecting pancreatic lesions in small pancreatic tumors. In our study, EUS also had high sensitivity for detecting stage 0 and stage I pancreatic lesions. EUS should be recommended for MPD dilation or stenosis cases, especially tumors that cannot be identified on CT scan and MRI.

However, we also have to recognize the limitations of EUS. The EUS sensitivity for small pancreatic tumors is higher than that of other modalities, but many cases cannot be diagnosed by EUS alone. In addition, the accuracy of EUS varies according to the skill of the operator. EUS should be performed by an experienced endoscopist who is proficient in the EUS technique, however, EUS alone may not always accurately differentiate malignancy from inflammation. In such cases, histologic examination using EUS–FNA will help to confirm the diagnosis.

### 4.2. Diagnosis of Early-Stage PC

Recently, EUS–FNA has been the standard method for the tissue diagnosis of PC. Meta-analyses have reported the sensitivity and specificity of EUS–FNA for PC as 85–92% and 96–98%, respectively [17,18,19,20]. For small pancreatic lesions, Sugiura et al. assessed the accuracy of EUS–FNA in 788 solid pancreatic lesions. The sensitivity for lesions less than 10 mm (*n* = 36) was 89.3%, for 10–20 mm lesions (*n* = 223) it was 95%, and for 20–30 mm lesions (*n* = 304) it was 97.4%. The sensitivity significantly increased as the lesion size increased [21]. Uehara et al. reported that the accuracy of EUS–FNA for pancreatic tumors smaller than 10 mm was 96% (*n* = 23) [22], and Takagi et al. reported that the accuracy of tumors smaller than 10 mm was 93% (*n* = 14) [23]. 

In our study, 12 patients underwent EUS–FNA in stage I, and the sensitivity was 83.3% (10/12). Two negative cases by EUS–FNA underwent pancreatic juice cytology and were diagnosed as positive for adenocarcinoma. For stage 0, although EUS detected tumor lesions in some cases, they were too small for EUS–FNA, and pancreatic juice cytology was performed; the sensitivity was 87.5%. These results suggest that EUS–FNA is useful for diagnosing small PCs; however, it cannot be used for pancreatic cancers that do not have sufficient mass lesions, including PCIS. In such cases, ERCP-based pancreatic juice cytology may be helpful for pathological diagnosis [24,25,26,27]. 

### 4.3. EUS Imaging Findings and Pathological Features of Stage 0 PC

We detected tumor lesions in 10 cases of stage 0 PC. Hypoechoic round lesions ranging from 10 to 20 mm in diameter were detected around the stenosed pancreatic ducts. Izumi et al. had investigated the EUS imaging findings and pathological features of 16 PCIS cases in Onomichi General Hospital. Nine cases had hypoechoic areas surrounding the MPD stenosis and for the pathological features, 81.3% (13/16) had chronic pancreatitis, 31.3% (5/16) had pancreatic fatty infiltration and all had subepithelial inflammatory cell infiltration and fibrosis in pancreatic parenchyma around the MPD stenosis [28]. Kikuyama et al. reported the pathological findings of stage 0 PC with a high degree of fatty changes in the pancreatic paraenchyma around the PCIS [29]. The JEDPAC also reported that local fatty changes may be specific to early-stage PC [16]. Hypoechoic lesions surrounding MPD stenosis is thought to reflect localized inflammation, fibrosis, and fatty changes due to PCIS [28,29,30]. Therefore, EUS–FNA cannot collect tissue in such cases and pancreatic juice cytology is useful for PCIS diagnosis.

## 5. Conclusions

The higher imaging resolution provided by EUS enables the detection of small pancreatic neoplasms. MPD dilation or MPD stenosis cases, especially tumors, cannot be identified on CT scan or MRI, therefore, EUS should be used. However, EUS has limitations, and in some cases, early PC cannot be detected by EUS. In such cases, ERCP-based cytology may be helpful for the diagnosis.

## Figures and Tables

**Figure 1 diagnostics-10-01086-f001:**
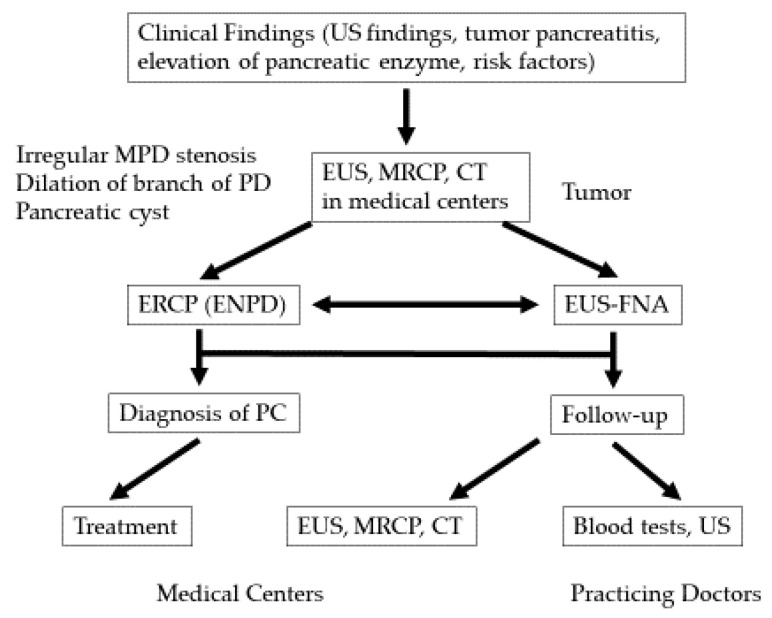
The algorithm for the diagnosis of pancreatic cancer (PC). CT: computed tomography, ENPD: endoscopic nasopancreatic drainage, ERCP: endoscopic retrograde cholangiopancreatography, EUS: endoscopic ultrasonography, FNA: fine-needle aspiration, MPD: main pancreatic duct, MRCP: magnetic resonance cholangiopancreatography, PD: pancreatic duct, US: ultrasonography.

**Figure 2 diagnostics-10-01086-f002:**
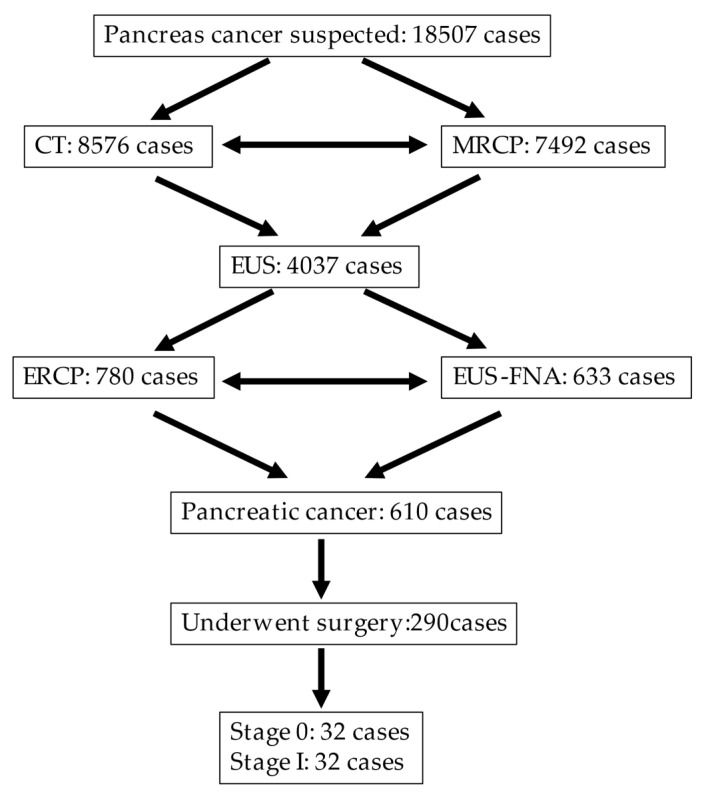
The flow of patient information. ENPD: endoscopic nasopancreatic drainage, ERCP: endoscopic retrograde cholangiopancreatography, EUS: endoscopic ultrasonography, FNA: fine-needle aspiration, MRCP: magnetic resonance cholangiopancreatography.

**Figure 3 diagnostics-10-01086-f003:**
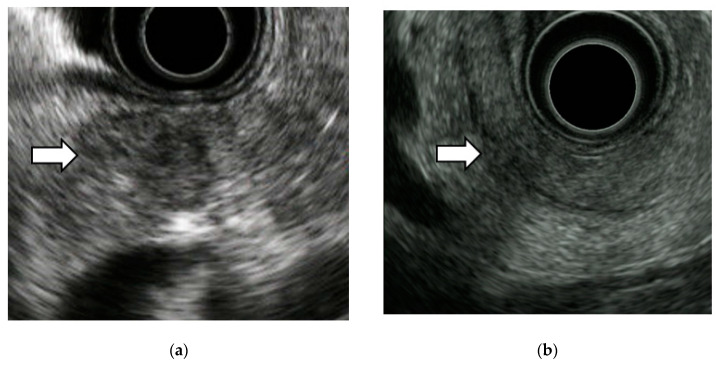
EUS imaging findings of stage 0 PC. Ten cases of stage 0 PC had hypoechoic lesions around MPD stenosis. Six cases showed a well circumscribed hypoechoic lesion (**a**). The other four cases showed a pale hypoechoic lesion with a relatively circumscribed lesion (**b**).

**Figure 4 diagnostics-10-01086-f004:**
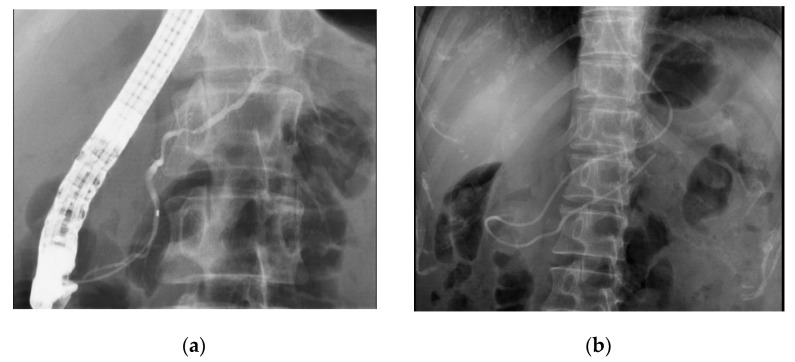
Pancreatic juice cytology. The guidewire was inserted to the stenosis of the pancreas duct through an ERCP (**a**). Then the wire remains and the ENPD catheter is placed into the pancreas duct (**b**). Pancreatic juice can be collected from an ENPD tube up to six times for 1 day.

**Table 1 diagnostics-10-01086-t001:** Risk factors for pancreatic cancer.

Family history
Pancreatic cancer
Hereditary pancreatic cancer syndrome
Accompanying diseases
Diabetes mellitus
Obesity
Chronic pancreatitis
Hereditary pancreatitis
Intraductal papillary mucinous neoplasm
Pancreatic cyst
Habits
Tobacco use
Heavy drinking

**Table 2 diagnostics-10-01086-t002:** The clinical characteristics of stage 0 and stage I PC.

	All Cases (*n* = 64)	Stage 0 (*n* = 32)	Stage I (*n* = 32)
Sex (male/female)	33/31	19/13	14/18
Age, mean (range)	71.3 (38–87)	71.3 (52–87)	71.8 (39–84)
Observation period (year), median (range)	3.9 (0.5–12.7)	4.2 (1.7–12.7)	3.4 (0.5–10.7)
Location, head/body/tail, n	25/32/7	11/16/5	14/16/2
1 year survival rate (%)		100	96.4
5 year survival rate (%)		78.9	66.7

**Table 3 diagnostics-10-01086-t003:** Imaging findings of stage 0 and stage I PC.

		All Cases, n (%)	Stage 0, n (%)	Stage I, n (%)
EUS		44/64 (68.8)	22/32 (68.8)	22/32 (68.8)
	MPD stenosis	33/44 (75)	16/22 (72.7)	17/22 (77.3)
	MPD dilation	33/44 (75)	17/22 (77.3)	16/22 (72.7)
	Tumor lesion	28/44 (63.6)	10/22 (45.5)	18/22 (81.8)
CT		58/64 (90.6)	31/32 (96.9)	27/32 (84.4)
	MPD stenosis	29/58 (50)	14/31 (45.2)	15/27 (55.6)
	MPD dilation	37/58 (63.8)	19/31 (61.3)	18/27 (66.7)
	Tumor lesion	18/58 (31)	3/31 (9.7)	15/27 (63)
MRI		54/64 (84.4)	31/32 (96.9)	23/32 (71.9)
	MPD stenosis	33/54 (61.1)	18/31 (58.1)	15/23 (68.2)
	MPD dilation	38/54 (70.4)	20/31 (64.5)	18/23 (78.3)
	Tumor lesion	12/54 (22.2)	3/31 (9.7)	9/23 (39.1)

MPD stenosis was detected in many cases, although tumors were seldom detected by CT and MRI, especially in stage 0. On the other hand, EUS had a high detection rate for tumor lesions in stage 0; 11 cases were not identified on a CT scan and MRI. EUS had a higher sensitivity for detecting early-stage PC than other modalities, although the sensitivity for stage 0 was 45.5%. There should be a limitation in detecting any tumors.

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
