# Peer review of "Endoscopic Ultrasonography Diagnosis of Early Pancreatic Cancerâ€"

_diagnostics, 2020, doi:10.3390/diagnostics10121086_

Round 1
Reviewer 1 Report
General comment: The authors presented an interesting work concerning to diagnostic of early pancreatic cancer by endoscopic ultrasonography.
Comments to the Author:
The manuscript should be revised by an English native.
Although interesting, recently a similar paper was published:
Endoscopic Ultrasound for Early Diagnosis of Pancreatic Cancer. Diagnostics (Basel). 2019 Sep; 9(3): 81.
doi: 10.3390/diagnostics9030081
Title: The title is short, concise and adequate.
Abstract: The full name should be provided before all abbreviations.
Introduction: The Introduction is adequate.
Materials and methods: In a general way, the methods are adequately described.
Was the patients’ consent obtained? Please clarify.
Results: They are clearly presented. The Figures and Tables are adequate.
Discussion: It is adequate.
Recommendation: The manuscript should be accepted for publication after a Minor revision.

Reviewer 2 Report
This study investigates an interesting topic. However, several methodological issues could be raised:
Please state if this is a prospective study. If yes, please provide Ethic Committee approval details.
Please provide a flow-chart of the study including how patients were selected, how many were excluded, and why.
Please provide details about EUS procedure including endosonographers experience.
Why EUS was not performed on all patients? Which were the criteria to perform or not perform EUS?
How the diagnosis of PDAC was confirmed? How many patients underwent resection?
Did you perform CH-EUS to evaluate the lesions?
Did you exclude patients with previous acute pancreatitis? False-positive cytology could be detected in inflammatory lesions, indeed.
To date, EUS technique for sampling pancreatic solid lesions is moving from EUS-FNA to EUS-FNB. Newest generation needles showed high histological yield (e.g., Di Leo M, et al. Dig Liver Dis. 2019 Sep;51(9):1275-1280; Crinò SF, et al. Gastrointest Endosc. 2020 Sep;92(3):648-658.e2.) This point could be of help in case of inconclusive EUS-FNA, rather than pancreatic juice cytology. Please discuss.
Author Response
Reviewer 2
This study investigates an interesting topic. However, several methodological issues could be raised:
Please state if this is a prospective study. If yes, please provide Ethic Committee approval details.
→This was retrospective study and approved by ethics commission at Onomichi General Hospital. P1 L16
“This study was approved by the ethics committees of Onomichi General Hospital (OJH-202043).”
Please provide a flow-chart of the study including how patients were selected, how many were excluded, and why.
→Added the flow-chart in Fig 1 and 2, and also added some information in 2.2. Patient Investigations. P4 L19-34
“We conducted a retrospective study, including patients who had undergone surgery and were diagnosed with PC from January 2007 to January 2020 at Onomichi General Hospital. The number of patients who consulted for a suspected PC was about 15000 cases, while that of patients who underwent CT was about 8500. Additionally, about 6000 patients who had main pancreatic duct (MPD) dilation underwent MRCP. Based on the CT and MRCP results, tumor lesions, irregular MPD stenosis, pancreatic duct branch dilation, and pancreatic cyst were detected; thus, we performed EUS on 4037 patients. If a tumor lesion was detected and had a considerable size for fine-needle-aspiration (FNA), we performed EUS-FNA to diagnose 633 patients. In cases of negative FNA, tumor lesion not detected with EUS, or lesions too small for FNA, we performed endoscopic retrograde cholangiopancreatography (ERCP) for pancreatic juice cytology on 780 cases. A total of 610 cases were histologically diagnosed with PC, while 290 underwent surgery. Sixty-four cases of early-stage PC were investigated postoperatively, consisting of 32 cases with stage 0 and 32 cases with stage I (Fig 2). We performed EUS for all suspicious PC cases. However, EUS was not performed in patients who refused or cases before 2010 when EUS was not commonly performed in hospitals.”
Please provide details about EUS procedure including endosonographers experience.
→Added the information. P5 L1-3
“EUS was performed by experienced specialists with more than 5 years of working experience and performed an average of more than 200 procedures every year.”
Why EUS was not performed on all patients? Which were the criteria to perform or not perform EUS?
→Added the information. P4 L32-34
“We performed EUS for all suspicious PC cases. However, EUS was not performed in patients who refused or cases before 2010 when EUS was not commonly performed in hospitals.”
How the diagnosis of PDAC was confirmed? How many patients underwent resection?
→Basically diagnosed histologically with EUS or ERCP and 290 patients had surgery. P4 L29-30
“A total of 610 cases were histologically diagnosed with PC, while 290 underwent surgery.”
→All Stage 0 and Stage I PC patients had surgery but some of them were cytology-negative postoperatively, but imaging findings cannot deny PC and surgery was performed. P8 L30-32
“Five patients with cytology-negative findings were suspected to have PC based on imaging results, and underwent surgery after providing consent; all cases were diagnosed as Stage 0 PC.”
Did you perform CH-EUS to evaluate the lesions?
→We don’t usually perform CH-EUS for pancreatic lesions because it is not applied for Japanese medical insurance.
Did you exclude patients with previous acute pancreatitis? False-positive cytology could be detected in inflammatory lesions, indeed.
→We didn’t exclude acute pancreatitis cases. We had not experienced false-positive cytology cases because of acute pancreatitis.
To date, EUS technique for sampling pancreatic solid lesions is moving from EUS-FNA to EUS-FNB. Newest generation needles showed high histological yield (e.g., Di Leo M, et al. Dig Liver Dis. 2019 Sep;51(9):1275-1280; Crinò SF, et al. Gastrointest Endosc. 2020 Sep;92(3):648-658.e2.) This point could be of help in case of inconclusive EUS-FNA, rather than pancreatic juice cytology. Please discuss.
→The new generation needles has high quality histologic samples and can improve the accuracy of EUS-FNA.
But, as I described in my study, EUS-FNA cannot be use for tumor that do not have sufficient mass (P10 L27-30), also the hypoechoic lesions around MPD stenosis of stage 0 PC is thought to reflect inflammation, fibrosis, and fatty changes(P11 L7-8), EUS-FNA or FNB cannot collect carcinoma tissue for these cases. We think that for stage 0 PC, pancreatic juice cytology is still useful for diagnosis.
I added a sentence in discussion. P11 L8-10
“Therefore, EUS-FNA cannot collect tissue in such cases and pancreatic juice cytology is useful for PCIS diagnosis.”
